# Inventory Model Design by Implementing New Parameters into the Deterministic Model Objective Function to Streamline Effectiveness Indicators of the Inventory Management

**Mária Stopková \*, Ondrej Stopka**  **and Vladimír Ľupták**

Department of Transport and Logistics, Faculty of Technology, Institute of Technology and Business in České Budějovice, 370 01 České Budějovice, Czech Republic
\* Correspondence: stopkova@mail.vstecb.cz

**Abstract:** The aim of this article is to modify the parameters and thus the objective function of the deterministic model of inventory theory so that other important aspects, which influence inventory management, can be taken into consideration. These aspects include the nature of inventory consumption, the share of inventories in sales, the capacity of means of transport and, above all, the reliability of suppliers. This goal is achieved by performing sophisticated and specific calculations for the individual parameters in the modified model. The modification of the objective function of the deterministic model has created a new multi-criteria model. The outcome of this model sought to optimize the supply process in a way that minimizes the risks associated with a lack of inventories while maintaining the economic effectiveness thereof. The model effectiveness is examined by comparing the application of the deterministic model and the proposed model with modified objective function. The results of applying these individual models have been produced based on calculations of indicators showing inventory management effectiveness—the speed of inventory turnover and the average number of inventories in storage.

**Keywords:** inventory theory; inventory model; deterministic model; inventory management performance indicators

## 1. Introduction

Models that apply methods of operational analysis are widely used for inventory management. As a matter of fact, an independent discipline that tackles the various economic and technical issues associated with material management, namely material/inventory theory, arose from operational analysis. Material theory (or more precisely the mathematical theory of inventory and production) deals with streamlining materials, semi-finished products, and product circulation, and the overall effectiveness of their management. The differences between models of material theory lie in which way and through which indicators the estimated consumption is defined, which probability model tests the consumption and the ordering cycle, and in which way the potential risks are estimated [1,2].

The deterministic model of inventory management is a basic model in the field of material theory. The parameters for the objective function of the model and its internal structure are mainly focused on the economics of inventory management. Nevertheless, this model takes into consideration only basic factors such as the cost associated with the supply process, the consumption intensity of particular inventory items and the delivery [3,4].

The aim of this article is to modify the objective function of the deterministic model of inventory theory and to make specific calculations for individual parameters in such a way that inventories

are optimized not only with respect to costs but also in regard to significant factors affecting the supply process [5]. These factors are the capacity of the means of transport, consumption, the share of inventories in the sales of a company and, last but not the least, the reliability of suppliers.

## 2. Literature Review

As inventory management is currently a topical issue, it has been addressed in many publications focusing on a variety of problems. It arises, in particular, from the need of enterprises for inventory management results to ensure certain market competitiveness. For instance, this matter is addressed by Atnafu and Balda [6] in their publication. They focused on small enterprises in Ethiopia and searched for empirical evidence in regard to inventory management that has a significant impact on the competitiveness of micro and small businesses. Inventory management efficiency is often measured through inventory management efficiency indicators. Specifically, it is the stock (inventory) turnover indicator that has been investigated in multiple publications [7–11]. The research study [7] deals with an empirical analysis of the impact of demand uncertainty in the US retail sector on stock turnover performance. The issue of unequal customer demand in the context of inventory is very topical as well. This topic is addressed by authors in the literature [8], where they search for an answer to the question of whether the high and low inventory turnovers of retailers respond differently to demand shocks?

Other options to influence inventory turnover positively have brought various innovations in the field including the horizontal or vertical integration of supply chain links. These issues have been discussed, for example, in publications [9,10]. In publication [9], the authors generally focus on the role of innovation in terms of the performance of inventory turnover, whereby publication [10] specifically deals with the impact of vertical integration on inventory turnover and operational performance. Inventory management performance indicators are influenced by both customer demand as well as supplier activity. The literature [11] proposes a hybrid algorithm that represents a model of inventory management with suppliers aiming to maximize inventory turnover in the manufacturer's warehouse. One way to increase inventory turnover is to choose the appropriate inventory management model. Many inventory management models have already been designed and operated. A general overview of stochastic models of inventory management is provided in publication [12], where the authors reviewed literature sources regarding stochastic models. In research study [13], the authors analyzed the performance of inventory management systems not only through the stochastic model but also from the deterministic model perspective. Deterministic models, or inventory-oriented models with deterministic demand, are addressed in publications [14–17] as well.

Different points of views on inventory model designs are examined in studies [18–21]. In publications [18,19], inventory models are proposed in such a way that various variabilities within inventory management are taken into account, in particular, whether variability is represented by delivery time [18], or the capacity utilization [19]. The general objective of inventory management is represented by the requirement for a minimum order size, which is addressed in publication [20], and also, an important aspect of any industrial area at present, environmental requirement is addressed in the literature [21]. Other models have been addressed in studies [22–26]. Those publications focused on various aspects of inventory management. The lost-sale model is analyzed in [22], the inventors' selection model in [23], an inventory model designed to efficiently redistribute production to individual orders (multi-objective model) is investigated in [24], a customer service model dealing with the service focus differentiation and response to time guarantees (a base-stock inventory model) in [25], and, of course, cost models of inventory management (continuous inventory models of diffusion type) are addressed in [26].

Publication [27] discusses the topic regarding the production process with supply issues and their cost performance (production–inventory system). Several publications, such as [28–34], address performance measurement in various fields in relation to inventory management. The impact of inventory management on overall supply chain performance is described by authors in publications [28–31]. While literature [30] examines the impact of retailers with knowledge of

supplier's inventory on supply chain performance, research study [31] investigates the effect of control system structure and the performance of an inventory goods flow system with long-variable delay. The interdependence among inventory types and firm performance is elaborated in publication [32]. The authors of literature reference [33] discuss financial performance assessment in terms of inventory management practices, and study [34] is aimed at performance assessment in homogeneous/heterogeneous collaborative enterprise networks with inventory adjustment.

The literature review conducted suggests that there are a number of research studies aimed at designing different inventory management models as well as evaluating inventory management performance. The objective of our manuscript is, based on a performed research study, to demonstrate the positive impact of a purposeful change in the objective function parameters on inventory management performance indicators.

## 3. Materials and Methods

The reliable delimitation of the optimal order quantity $Q_{opt}$ depends on multiple factors. For the purpose of this research study, the existing deterministic model represents the principle based on which a new inventory model can be proposed, whereby the relationships of factors affecting the whole inventory management are investigated. Analyzing the deterministic model objective function of inventory theory is the subject of the Section 3.1.

### 3.1. Delimitation of the Optimal Order Quantity by the Deterministic Model

Under the deterministic model, the order size $Q$ and order cycle $t$ have a steady character. In such cases, it is possible for an order to be available when the very last item is out of stock. However, problems arise when a particular stock item is consumed intermittently, or its consumption fluctuates. This implies that the use of the deterministic model for inventory management is therefore confined to steadily-consumed stock items [35–37].

As a result, the application of the deterministic model of inventory management requires an XYZ analysis. The objective function of the deterministic model is applied when the focus is on the overall annual costs of the supply process $N^C$, whereby the following parameters are taken into consideration:

- delivery cost $N_{dod}$;
- storage cost $n_{skl}$;
- purchase price $C^N$.

The aim of the deterministic model is to minimize the objective function (1):

$$(Q) = N_{dod} + n_{skl} + C^N \rightarrow min \tag{1}$$

where: $N^C$—overall costs [money]; $N_{dod}$—delivery cost [money]; $n_{skl}$—storage cost [money]; $C^N$—purchase price [money] [35].

The objective function of the deterministic model focuses on minimizing the cost associated with supplies, i.e., it only deals with the economics of inventory management. Delivery cost $N_{dod}$ in the deterministic model includes all cost items related to obtaining the supplies (orders, phone fees, fax, postage, and transport cost) associated with one order, which are subject to change [36].

The main purpose of the deterministic model of inventory management is to optimize the order size $Q$ and the inventory level $H$, while taking into consideration the economic criterion of the overall annual costs of the supply process $N^C$ [35,38]. The outcome of the application of the deterministic model discloses information about:

(a)	order size $Q_{opt}$ (see Equation (2))

$$Q_{opt} = \sqrt{\frac{2 \times I \times N_{dod}}{n_{skl}}} \ [\text{m.u.}] \tag{2}$$

where: $I$—consumption intensity per day [unit quantity]; $N_{dod}$—delivery cost [money]; $n_{skl}$—storage cost [money] [35].

(b) order cycle duration $t$ (see Equation (3))

$$t = \frac{Q_{opt}}{I} \times 365 \ [\text{days}] \tag{3}$$

(c) inventory level $H$ (see Equation (4)–(6))

If $d \leq t$; then

$$H = d \times \frac{I}{365} \ [\text{m.u.}] \tag{4}$$

where: $d$—delivery time [days].

If $d > t$; then

$$H = d \times \frac{I}{365} - m \times Q_{opt} \ [\text{m.u.}] \tag{5}$$

$$m = \frac{d}{t} \tag{6}$$

Based on knowledge of the deterministic model of inventory management and comprehending the fundamental principles of its conception as well as its internal logic, the new inventory management model was designed. Section 3.2 is particularly devoted to proposal of that model objective function.

### 3.2. Proposing the New Objective Function of the Inventory Management Model

The modified objective function is not confined to only minimizing the cost associated with inventories, but also on the extent to which inventory funds are related to the minimization of the risks associated with a lack of inventories. In order to specify the objective function, the degree of risk associated with the lack of inventories is calculated as the potential losses a company would suffer as a result of having a certain lack of inventories $S_{ned}$. The parameters for the objective function of the modified model are:

- annual delivery cost $N_{dod}{}^{R}$;
- inventory cost and their dependence on inventory funds $W$;
- losses incurred due to a lack of inventories $S_{ned}$.

The aim of the modified model is to minimize the objective function (see Equation (7)):

$$N^{C}(Q) = N_{dod}^{R} + W + S_{ned} \rightarrow min \tag{7}$$

where: $N^{C}$—overall costs [money]; $N_{dod}{}^{R}$—annual delivery cost [money]; $W$—dependence on inventory funds [money]; $S_{ned}$—losses incurred due to lack of inventories [money].

Since the modified objective function includes the potential losses that would be incurred due to the lack of inventories $S_{ned}$, the modified model is no longer only focused on minimizing costs, but also takes into consideration other strategic criteria, such as the share of inventories in sales $F$ and the reliability of suppliers $DS$.

The following sections (Sections 3.2.1–3.2.3) specify methods for calculating the individual parameters for the modified objective function.

### 3.2.1. Delivery Cost Determination Method

The optimization of inventories using the modified model takes into consideration annual delivery cost $N_{dod}{}^{R}$. The bigger the offer, the higher the delivery costs are (upon exceeding the capacity of the transport unit $k^{PJ}$). In contrast, if the amount of annual orders $R$ decreases by increasing the order size $Q^{N}$, the annual delivery cost $N_{dod}{}^{R}$ decreases—as is suggested in Figure 1. When applying the new

parameter $N_{dod}{}^R$, the modified model takes into consideration the following factors that influence the management process:

- capacity of transport unit $k^{PJ}$;
- consumption $R$; $r$;
- consumption intensity $I$;
- knowledge that delivery cost $N_{dod}$ considerably changes upon exceeding the capacity;
- specific order size $N_{dod}{}^{PJ}$.

The calculation of the annual delivery cost $N_{dod}{}^R$ for the purposes of the application of the model with modified objective function is as follows (see Equation (8)):

$$N^R_{dod} = \frac{r}{Q^N/I} \times R \times N^{PJ}_{dod} \times \frac{Q^N}{k^{PJ}} \qquad (8)$$

where: $N_{dod}{}^R$—annual delivery cost [money]; $r$—consumption period [days]; $Q^N$—order size [unit quantity]; $I$—consumption intensity per day [unit quantity]; $R$—number of consumption periods per year [-]; $N_{dod}{}^{PJ}$—cost of delivery of one transport unit [money]; $k^{PJ}$—capacity of transport unit [unit quantity]; $k^S$—storage capacity [unit quantity].

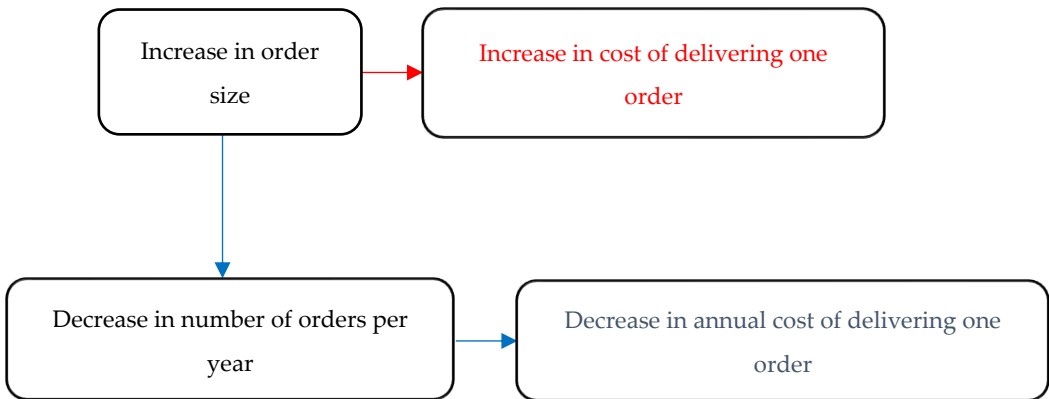

**Figure 1.** Dependence of delivery cost on order size. Source: Authors.

Figure 1 depicts the relationship between factors affecting the annual delivery cost $N_{dod}{}^R$. The definition of this relationship is necessary for the relevant specification of inventories of $N_{dod}{}^R$ calculation and the subsequent optimization of inventories by minimizing the proposed objective function.

### 3.2.2. Storage Cost and Purchase Price Determination Method

The objective function of the modified model, together with the objective function of the deterministic model, includes $n_{skl}$; however, this parameter is an integral part of the specific calculation of parameter $W$, which, apart from $n_{skl}$, is reflected in the extent to which the funds are related to inventories $VFP$.

Both parts of parameter $W$ ($n_{skl} + VFP$) depend on the inventory level $H$ maintained in the company, the level of which indicates the consumption intensity of inventories $I$, the delivery period $d$ and also the reliability of the particular supplier $DS$. The incorporation of the $DS$ parameter in the objective function considerably reduces the risks associated with the lack of inventories, and thereby ensures standard customer service, while the economic effectiveness of the deterministic model is not omitted. The $VFP$ parameter depends on the purchase price of the specific inventory $C^N$, which refers to another aspect of the objective function of the deterministic model. Parameter $W$ in the modified model takes the following into consideration:

- consumption intensity *I*;
- delivery period *d*;
- reliability of the supplier *DS*;
- purchase price $C^N$.

The calculation of parameter *W* for the application of the model with modified objective function is as follows (see Equation (9)):

$$W^H = \left(I \times d + Q^N \times DS\right) \times \left(n_{skl} + C^N\right) \tag{9}$$

where: $W^H$—capital tied up in inventory [money]; *I*—consumption intensity per day [unit quantity]; *d*—delivery period [days]; $Q^N$—order size [unit quantity]; *DS*—reliability of supplier [-]; $n_{skl}$—storage cost [money]; $C^N$—purchase price [money].

3.2.3. Risk of Lack of Inventories Determination Method

The objective function of the modified model of inventory management not only focuses on previous cost values, but also on minimizing the risks associated with the lack of inventories, as defined by the parameter $S_{ned}$. The value of this parameter depends on the following factors:

- reliability of supplier *DS*;
- share of inventories in sales *F*;
- purchase price $C^N$;
- consumption intensity *I*;
- delivery period *d*.

The calculation for the losses associated with the lack of inventories $S_{ned}$ for the application of the model with modified objective function is as follows (see Equation (10)).

$$S_{ned}^N = Q_{opt} \times \left(\frac{F}{100} \times C_p\right) - \left(I \times d + Q^N \times DS\right) \times \left(\frac{F}{100} \times C_p\right). \tag{10}$$

where: $S_{ned}{}^N$—losses due to lack of inventories [money]; $Q_{opt}$—optimal order size [unit quantity]; *F*—percentage share in sales [%]; $C^P$—purchase price of final product [money]; *I*—consumption intensity per day [money]; *d*—delivery period [days]; $Q^N$—order size [unit quantity]; *DS*—reliability of supplier [-].

Based on the proposed objective function, specific indicators, as follow, can be calculated:

(a)    order size $Q_{opt}$;

For inventories which are consumed continuously or with less fluctuation, Equation (11) is applied:

$$Q_{opt} = \sqrt{\frac{2 \times I \times N_{dod}^R}{W}} \; [\text{m.u.}] \tag{11}$$

For inventories which are consumed randomly, Equation (12) is applied:

$$Q_{opt} = I \times d \; [\text{m.u.}] \tag{12}$$

(b)    order cycle duration *t* (see Equation (13));

$$t = \frac{Q_{opt}}{I} \; [\text{days}] \tag{13}$$

(c)  inventory level $H$ (see Equation (14));

$$H = \sqrt{\frac{2 \times S_{ned}}{W}} \ [\text{m.u.}] \tag{14}$$

3.2.4. Designation of Research Prerequisites

The subject of this research study will also be to either confirm or disprove the designated research prerequisites as follows:

- P1: Incorporation of parameter $N_{dod}{}^R$ in the objective function of the modified model does not increase the average inventory level $\overline{Q}$.
- P2: Incorporation of parameter $N_{dod}{}^R$ in the objective function of the modified model does not decrease the speed of inventory turnover $v$.
- P3: Incorporation of parameter $W$ in the objective function of the modified model does not increase the maintained inventory level $H$ in the company.
- P4: Incorporation of parameter $W$ in the objective function of the modified model does not decrease the speed of inventory turnover $v$.
- P5: Incorporation of parameter $S_{ned}$ in the objective function of the modified model does not increase the maintained inventory level $H$ in the company.

Given research prerequisites have been formulated based on the objectives stipulated during designing the new inventory management model. Confirming or disproving the specified prerequisites will be undertaken through a particular case study, in which the application results of the deterministic and proposed model will be compared (see Section 4).

## 4. Comparison of Applying the Deterministic Model and the Designed Model

A research study dealing with the optimization of the supply process was used to compare the application of the deterministic model and that of the model with modified objective function. Three representatives in relation to inventories were chosen in a specific company according to their stochastic character in the consumption of the particular inventory item. The input data is presented in Table 1.

**Table 1.** Input data.

| Parameter | Abb. | Unit of Measure | Inventory | | |
|---|---|---|---|---|---|
| | | | $\alpha$ | $\beta$ | $\gamma$ |
| Delivery cost of one order | $N_{dod}$ | [€] | 0.5 | 0.80 | 0.5 |
| Unit storage cost | $n_{skl}$ | [€] | 0.15 | 0.12 | 0.15 |
| Capacity of transport unit | $k^{PJ}$ | [m.u.] | 20 pcs | 11 pcs | 200 kg |
| Delivery period | $d$ | [days] | 3 pcs | 2 pcs | 1 kg |
| Consumption intensity | $I$ | [m.u./year] | 1204.5 pcs | 2920.0 pcs | 182.5 kg |
| | | [m.u./day] | 3.3 pcs | 8.0 pcs | 0.5 kg |
| Purchase price | $C^N$ | [€] | 100 | 20 | 2 |
| Consumption period | $r$ | [days] | 365 | 60 | 20 |
| Number of consumption periods | $R$ | [-] | 1 | 2 | 0 |
| Reliability of supplier | $DS$ | [-] | 0.95 | 0.85 | 0.9 |
| Purchase price | $C^P$ | [€] | 220 | 180 | 150 |
| Share of inventories in sales | $F$ | [%] | 95 | 60 | 5 |

Source: Internal data of the selected company.

Following the data on specific inventory items of the given company, the calculations needed to optimize the supply process using the deterministic model (Section 4.1) and the proposed model (Section 4.2) will be performed in following sections.

### 4.1. Determination of the Optimal Order Size

The extent and parameters for the calculation of the optimal order size $Q_{opt}$ differ for the model with modified objective function and the deterministic model. In contrast to the deterministic model, the modified model of inventory management takes into consideration the daily consumption of inventories $I$ [m.u./day] in contrast to the annual consumption of inventories $I$ [m.u./year] and the annual delivery cost $N_{dod}{}^R$ in contrast to one-off delivery cost $N_{dod}$.

The reason for calculating annual delivery cost is that this value also takes into consideration the number of orders carried out per year $R$, which depends on order size $Q$; on the one hand, the smaller the order, the lower the one-off delivery cost is $N_{dod}$, while, on the other hand, it means a larger number of orders carried out per year $R$, which also means an increase in delivery cost per year $N_{dod}{}^R$.

Furthermore, the calculation of annual delivery cost $N_{dod}{}^R$ must take into consideration changes in delivery cost as a result of exceeding the capacity of the means of transport $k^{PJ}$ and the consumption of the particular inventory, which in turn significantly influences the number of orders carried out per year $R$, and which is closely associated with annual delivery cost $N_{dod}{}^R$.

Another change regarding the parameters consists in the calculation of the optimal order size $Q_{opt}$ according to the modified model and its dependence on the newly created parameter $W$, which substitutes individual storage cost $n_{skl}$ in the deterministic model. Parameter $W$, apart from storage cost $n_{skl}$, also takes into consideration their dependence on inventory funds $VFP$, which enhances the economic effectiveness of inventory management. The inventory level $H$ means the key variable for the calculation of parameter $W$ depends on consumption intensity $I$, delivery period $d$, purchase price $C^N$ and the reliability of the supplier $DS$. This not only increases economic effectiveness, but also minimizes the risks associated with the lack of inventories.

Figure 2 illustrates how the optimal order size is determined through distribution curves that indicate changes in the $N_{dod}{}^R$ and $W$ parameters in relation to order size $Q^N$ for the model with modified objective function.

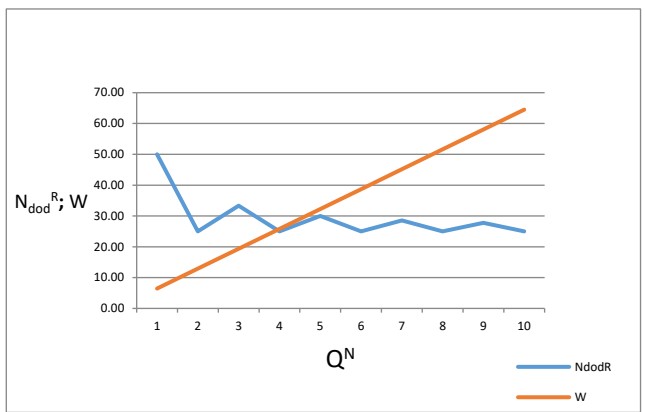

**Figure 2.** Determination of the optimal order size.

The blue curve represents the change in $N_{dod}{}^R$ depending on $Q^N$ at the limited vehicle capacity. The red curve depicts the dependence of the capital tied up in inventory $W$ on $Q^N$ at a certain intensity and the nature of the given inventory consumption, while also taking into account the delivery period and reliability of the supplier in terms of the delivery process.

Table 2 compares calculations for the optimal order size $Q_{opt}$ according to the deterministic model of inventory management and the model with modified objective function. It also includes specific calculations for the optimal order size $Q_{opt}$ for the individual representatives of the inventories ($\alpha$, $\beta$, $\gamma$) identified and defined in Table 1.

**Table 2.** Calculations for optimal order size according to both models.

| Deterministic Model * | | | Proposed Model ** | | |
|---|---|---|---|---|---|
| According to Equation (2) | | | According to Equation (11) | | According to Equation (12) |
| $\alpha$ * | $\beta$ * | $\gamma$ * | $\alpha$ ** | $\beta$ ** | $\gamma$ ** |
| 90 pcs | 197 pcs | 35 kg | 14 pcs | 6 pcs | 0.5 kg |

Explanatory notes: *—calculation according to the deterministic model; **—calculation according to the proposed model.

### 4.2. Determination of the Order Cycle Duration

The order cycle duration t depends on the optimal order size $Q_{opt}$ and the consumption intensity *I*. The deterministic model and the model with modified objective function differ in the way they determine the order cycle *t*, whereby, under the former, the calculation is based on annual consumption *I* [m.u./year] and, under the latter, on daily consumption *I* [m.u./day].

The order cycle duration t for the modified model is calculated only for inventories consumed continuously. Orders for other inventory items are classified according to 'inventory level' *H* (according to Equation (14)). The calculations for the order cycle duration on the basis of both models are presented in Table 3.

**Table 3.** Calculations for order cycle according to both models.

| Deterministic Model * | | | Proposed Model ** | | |
|---|---|---|---|---|---|
| According to Equation (3) | | | According to Equation (13) | | |
| $\alpha$ * | $\beta$ * | $\gamma$ * | $\alpha$ ** | $\beta$ ** | $\gamma$ ** |
| 27 days | 25 days | 70 days | 4 days | 0.75 day | 1 day |

Explanatory notes: *—calculation according to the deterministic model; **—calculation according to the proposed model.

The lower the optimal order size $Q_{opt}$ derived from the modified model of inventory management, the shorter the order cycle *t*; this fact has a favorable impact on the indicator for the speed of inventory turnover *v*. The higher the speed of inventory turnover *v*, the more economically effective the inventory management.

### 4.3. Determination of the Inventory Level

The methods for calculating the inventory level *H* significantly differ between the two models. Under the deterministic model, the determination of inventory level *H* is based on consumption intensity *I*, the delivery period *d* and the order cycle duration *t*.

The modified model of inventory management determines the inventory level *H* in such a way that minimizes the likelihood of the risk of a lack of inventories to almost zero. The method for the calculation of the losses that would be suffered as a result of the lack of inventories $S_{ned}$ (10), allows other factors relevant to the supply process to be taken into consideration. These factors include the reliability of the supplier of the particular inventory *DS*, the extent to which the specific inventory participates in the sales of the final product F and the purchase price of the specific inventory $C^N$.

Figure 3 illustrates how the optimal inventory level is determined through distribution curves that indicate changes in the *W* and $S_{ned}$ parameters in relation to inventory level $H^N$ for the model with modified objective function.

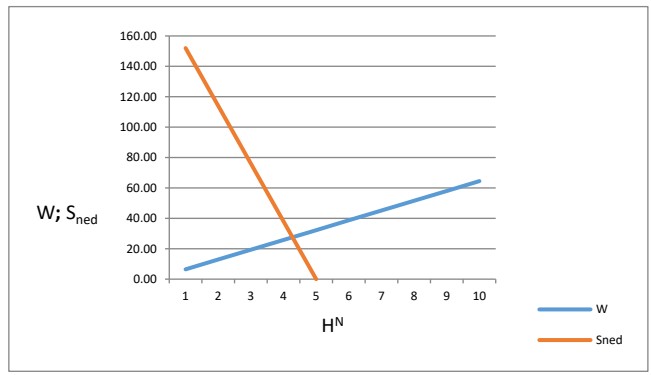

**Figure 3.** Determination of the optimal order size.

The blue curve represents the dependence of the capital tied up in inventory $W$ on $H^N$ at a certain intensity and the nature of given inventory consumption, while also taking into account the delivery period and reliability of the supplier in terms of the delivery process. The red curve shows the likelihood of the risk of a lack of inventories and the resulting cost $S_{ned}$ depending on the inventory level in the warehouse $H^N$.

The calculations for the emergency supply inventory $\alpha$ and the determination of the signal inventory level $H$ for inventory $\beta$ and $\gamma$ according to the deterministic and modified model of inventory management are presented in Table 4.

**Table 4.** Calculations for inventory level according to both models.

| Deterministic Model * | | | Proposed Model ** | | |
|---|---|---|---|---|---|
| According to Equations (4)–(6) | | | According to Equation (14) | | |
| $\alpha$ * | $\beta$ * | $\gamma$ * | $\alpha$ ** | $\beta$ ** | $\gamma$ ** |
| 89 pcs | 8 pcs | 0.5 kg | 4 pcs | 0 pcs | 2 kg |

Explanatory notes: *—calculation according to the deterministic model; **—calculation according to the proposed model.

The modified model of inventory management determines:

- the size of the emergency supply for inventories consumed continuously through the year;
- the size of the signal inventory for inventories with a consumption that is affected by seasonal fluctuations or surges.

## 5. Results and Discussion

In general, most of the published studies dealing with inventory management issues are aimed at applying existing inventory management models (see literature review chapter) as well as publications [38–44]. Unlike those literature sources, this manuscript is focused on an in-depth analysis of the existing model internal structure, i.e., the deterministic model of inventory management, and investigates options to improve its objective function through the appropriate design of its parameters. The research purpose of this article is, thus, to propose the inventory management model that, when applied in practice, will allow a company to optimize the inventory level as well as the entire supply process, while taking into account the whole scale of criteria that have not been incorporated into existing models yet. These criteria do not only consist of a cost criterion, as is the case with existing models [12,45], but also consider other significant factors affecting supply process continuity.

In contrast to the parameters for the deterministic model of inventory management, the model with modified objective function not only optimizes inventories in relation to delivery cost $N_{dod}$, storage $n_{skl}$ and the purchase price of inventory $C^N$, but also takes into consideration the potential losses that would be suffered if inventories were not available on time $S_{ned}$. Moreover, the modified model works

with an annual delivery cost $N_{dod}{}^R$ instead of one-off delivery cost $N_{dod}$, which results in the long-term sustainable economic effectiveness of the supply process.

The incorporation of parameter $N_{dod}{}^R$ in the objective function of the modified model may cause a considerable increase in the average inventory level $\overline{Q}$, which would consequently reduce the speed of inventory turnover $v$. The reduction in the speed of inventory turnover $v$ along with the increase in the inventory level $H$ in the company represents emergency supplies, i.e., signal supply may be caused by incorporating parameter $W$ in the objective function of the modified model. Apart from parameter $W$, the incorporation of parameter $S_{ned}$ may also play an important role in increasing the maintained inventory level $H$ in the company.

The average inventory level $\overline{Q}$, together with the speed of inventory turnover $v$, is a leading indicator of the effectiveness of inventory management [39]. The average inventory level $\overline{Q}$ amounts to $\frac{1}{2}$ the optimal order size $Q_{opt}$ (see Equation (15)).

$$\overline{Q} = \frac{Q_{opt}}{2} \ [\text{m.u.}] \tag{15}$$

Table 5 compares the figures for the average inventory levels for the individual items $\overline{Q}$ regarding the optimal order size $Q_{opt}$ according to both models of inventory management.

**Table 5.** Comparison of average inventory levels according to both models.

| Deterministic Model * | | | Proposed Model ** | | |
|---|---|---|---|---|---|
| | | According to Equation (15) | | | |
| $\alpha$ * | $\beta$ * | $\gamma$ * | $\alpha$ ** | $\beta$ ** | $\gamma$ ** |
| 45 pcs | 98.5 pcs | 17.5 kg | 7 pcs | 3 pcs | 0.25 kg |

Explanatory notes: *—calculation according to the deterministic model; **—calculation according to the proposed model.

Under the modified model of inventory management, the average inventory level $\overline{Q}$ for $\beta$ and $\gamma$ is significantly lower because, unlike the deterministic model, it takes fluctuations and surges in inventory consumption into consideration.

This important fact is further confirmed by the indicator for the speed of inventory turnover $v$, which shows how many times per year, i.e., reporting periods, the original inventories are changed for new ones (see Equation (16)). The key goal is to achieve the highest possible speed of inventory turnover $v$ [40].

$$v = \frac{1}{t} \times r \ [\text{times/year}] \tag{16}$$

In Table 6, the figures for the optimal speed of inventory turnover $v$ according to both models of inventory management are compared.

**Table 6.** Calculations for the optimal speed of inventory turnover.

| Deterministic Model * | | | Proposed Model ** | | |
|---|---|---|---|---|---|
| | | According to Equation (16) | | | |
| $\alpha$ * | $\beta$ * | $\gamma$ * | $\alpha$ ** | $\beta$ ** | $\gamma$ ** |
| 13.52 x/year | 2.4 x/year | 0.3 x/year | 91.25 x/year | 80 x/year | 20 x/year |

Explanatory notes: *—calculation according to the deterministic model; **—calculation according to the proposed model.

Table 7 compares the results of the application of the modified model (light gray areas) to that of the deterministic model (dark gray areas) of inventory management.

**Table 7.** Comparison of the results of the application of both inventory management models.

| | $Q_{opt}$ | | $t$ | | $H$ | | $\overline{Q}$ | | $v$ | |
|---|---|---|---|---|---|---|---|---|---|---|
| α | 90 pcs * | 14 pcs ** | 27 days * | 4 days ** | 89 pcs * | 4 pcs ** | 45 pcs * | 7 pcs ** | 13.52 x/year * | 91.25 x/year ** |
| β | 197 pcs * | 6 p pcs ** | 25 days * | 0.75 day ** | 8 pcs * | 0 pcs ** | 98.5 pcs * | 3 pcs ** | 2.4 x/year * | 80 x/year ** |
| γ | 35 kg * | 0.5 kg ** | 70 days * | 1 day ** | 0.5 kg * | 2 kg ** | 17.5 kg * | 0.25 kg ** | 0.3 x/year * | 20 x/year ** |

Explanatory notes: *—calculation according to the deterministic model; **—calculation according to the proposed model.

Based on the obtained results summarized in the table above (see Table 7), predesignated research prerequisites (see Section 3.2.4) may be confirmed or disproved. For better clarity, the research prerequisites, as well as their confirmation or disproving, are listed in the Table 8 as follows.

**Table 8.** Confirmation/disproving of the determined research prerequisites.

| | Research Prerequisites | Status |
|---|---|---|
| P1 | Incorporation of parameter $N_{dod}{}^R$ in the objective function of the modified model does not increase the average inventory level $\overline{Q}$. | confirmed |
| P2 | Incorporation of parameter $N_{dod}{}^R$ in the objective function of the modified model does not decrease the speed of inventory turnover $v$. | confirmed |
| P3 | Incorporation of parameter $W$ in the objective function of the modified model does not increase the maintained inventory level $H$ in the company. | confirmed |
| P4 | Incorporation of parameter $W$ in the objective function of the modified model does not decrease the speed of inventory turnover $v$. | confirmed |
| P5 | Incorporation of parameter $S_{ned}$ in the objective function of the modified model does not increase the maintained inventory level $H$ in the company. | confirmed |

Following the results of the case study conducted within this manuscript, it can be stated that the optimal order size is lower for all the inventory item types that have been the subject of the research, which results in a shorter order cycle as well. This is due to the fact that the proposed model, in addition to storage cost, also takes into account the level of capital tied up in inventory items, which are dependent on the inventory level in the warehouse. For the same reason, the optimal inventory level kept in the warehouse after the proposed model application is lower compared to the results of the deterministic model application and, thereby, the average inventory level in the warehouse is lower as well. These facts have a positive effect on the storage cost value as well as the overall level of capital tied up in inventory. Due to the lower optimal order size, the order cycle is also shorter, thus increasing the inventory turnover rate, which is a significant indicator of the supply process being managed efficiently.

## 6. Conclusions

The objective of this article was to modify the parameters and thus the objective function of the inventory theory deterministic model in order that other important aspects which influence inventory management could be taken into consideration. In doing so, account was not only taken of the cost associated with inventories, consumption intensity and delivery period, but also the capacity of the means of transport, consumption, the share of inventories in sales and, last but not the least, the reliability of suppliers.

The effectiveness of optimizing inventories by implementing the modified model was verified through its application and the subsequent comparison of the results with the existing deterministic model. Order size, order cycle duration and sustained inventory levels were compared, as were the speed of inventory turnover and the average inventory level as key indicators of the effectiveness of inventory management.

The results of the research study have revealed that the optimal order size is smaller for all the inventories that were included in the research study, which indicates a shorter order cycle. It results from the fact that inventories optimized through the model with modified objective function not only take into consideration storage cost, but also their dependence on inventory funds, which is dependent on the inventory level in storage.

For this reason, the optimal sustained inventory level in storage was lower after the application of the modified model in comparison to the results obtained through the deterministic model; the average inventory level in storage was therefore also lower. These observations have a favorable impact on the amount of storage cost and overall dependence on inventory funds. Furthermore, the smaller optimal order size indicates a shorter order cycle, which in turn means a higher speed of inventory turnover, which is a key indicator of an effectively controlled supply process.

As for future research recommendations, due to the wide range of criteria within the model proposed leading to large-scale calculations, it would be, therefore, reasonable to develop a software application based on the designed inventory model in order to automate the overall or partial inventory optimization processes. Another alternative may be to implement the proposed model in MS Excel using the specific mathematical, statistical and graphical functions. After entering the input parameters, the developed software application should allow calculating and creating optimization graphs to determine the optimal order size and the optimal inventory level, as well as the optimal order cycle, in an enterprise. Developing the specific SW application for the designed model would help towards automating optimization processes that would greatly contribute to both time and cost savings.

**Author Contributions:** O.S. and M.S. conceived and designed the manuscript; they elaborated its methodology, processed the literature review, verified the results and performed the experiments; V.Ľ. performed individual calculations, realized the final manuscript corrections and supervised the formal revision of the manuscript.

**Funding:** This research received no external funding.

**Acknowledgments:** This manuscript was supported within solving the research project entitled "Autonomous mobility in the context of regional development LTC19009" of the INTER-EXCELLENCE program, the VES 19 INTER-COST subprogram.

**Conflicts of Interest:** The authors declare no conflict of interest.

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
