# Peer review of "Inventory Model Design by Implementing New Parameters into the Deterministic Model Objective Function to Streamline Effectiveness Indicators of the Inventory Management"

_sustainability, doi:10.3390/su11154175_

Round 1
Reviewer 1 Report
An interesting article, although only theoretical. Is substantively correct. Applies to current issues. The article contains an extensive literature review. Correct analysis. Calculations carried out correctly. Appropriate applications. It's positives.
Suggestions for improvement:
1) In chapter 2.1 (Delimitation of the Optimum Order of Quantity by Deterministic Model) I would propose after the formula (6) to give the text of the ending of the subsection. Similarly in the case of sub-chapter 2.2.1. Finally, after the drawing should be a summary text of the content contained in 2.2.1. Analogously to table 1. Etc.
2) I suggest not to end chapters (eg 2.2.4) with bullets. Can you provide information about the origin of the source data? As before, there is no summary text of the content presented in 2.2.4 and the entire chapter 2.
3) I suggest you consider recall in the text in Figure 2 before placing it. First a reference in the text, then a drawing. This is the case with Fig. 1, Fig. 3 and, e.g., tab. 1.
4) The article is theoretical. Is it possible to present the implementation of the model in reality? If there are actual results, I would suggest presenting them.
Suggestions for improvement are discretionary. They are not obligatory.
Author Response
- bullets behind subchapters’ numbers has been removed,
- the particular text of the subsections’ endings has been added,
- in chapter 4, the way of proposed model implementation into real conditions has been added,
- in Table 1, information on the input data source origin has been entered,
- reference (recall) regarding individual figures in the text prior to them has been added.
Please, see the attachement as well.

Reviewer 2 Report
Generally I am satisfied of the quality level
Some suggestions:
- Decrease the Introduction and prepare "literature review" section as separate
Consider some more interesting sources:
Kot, S., Grondys, K., Szopa, R. Theory of inventory management based on demand forecasting (2011) Polish Journal of Management Studies, 3, pp. 148-156. and more from Scopus journals Improve the results discussion using some comparision to other sources.
Author Response
- the literature review part has been prepared as an individual chapter – as required by the reviewer,
- comparison with other studies has been added into the results and discussion chapter to specify particular distinctions between each other,
- the proposed literature reference by the reviewer has been added,
- even other references from the Scopus database have been added.
Please, see the attachment as well.

Reviewer 3 Report
On the conceptual level the authors try to develop a unifying theory for economic order quantity, and this is interesting.
However, on the mathematical level, this is not an easy task. Specifically, the authors provide equations for various cost terms that are not always clear. For example, in equation (9) a decrease in supplier reliability, decreases the dependence of inventories on fund. I cannot understand this dependency.
The conclusions are case dependent. If the authors want to generalize their findings, they should provide a mathematical proof that e.g. the order quantity of the new model is always less than the traditional EOQ.
Finally, there are no explanations for the figures. For example, the saw-teeth line in figure 2 could be explained through the capacity of the transportation mean. In such cases in real world, we employ alternative inventory control policies with fixed order quantity (equal to the truck capacity).
In conclusion, I believe that the manuscript needs extra effort from the authors to be suitable for publication.
Author Response
- the term “dependence of inventories on funds” has been replaced by the term “capital tied up in inventory”,
- the detailed description and clarification of the content of Figures 2 and 3 have been added, wherein, particularly, in Figure 2, the blue curve represents the change in NdodR depending on QN at the limited vehicle capacity, and the red curve depicts the dependence of the capital tied up in inventory W on QN at certain intensity and nature of the given inventory consumption, while also taking into account the delivery period and reliability of the supplier in terms of the delivery process. Whereas, in Figure 3, the blue curve represents the dependence of the capital tied up in inventory W on HN at certain intensity and nature of the given inventory consumption, while also taking into account the delivery period and reliability of the supplier in terms of the delivery process, and the red curve shows the likelihood of the risk of a lack of inventories and the resulting cost Sned depending on the inventory level in the warehouse HN.
- the detailed clarification and explanation of the facts regarding the declaration that the proposed model application leads to the lower optimal order quantity, decreased order cycle as well as minimized inventory level in the company, resulting in lower average inventory level in the warehouse and higher inventory turnover rate (in comparison with the deterministic model application) has been added. Specifically, following the results of the case study conducted within our manuscript, it can be stated that the optimal order size is lower for all the inventory item types that have been the subject of the research, which results in a shorter order cycle as well. This is due to the fact that the proposed model, in addition to storage cost, also takes into account the level of capital tied up in inventory items, which are dependent on the inventory level in the warehouse. For the same reason, the optimal inventory level kept in the warehouse after the proposed model application is lower compared to results of the deterministic model application; and thereby, the average inventory level in the warehouse is lower as well. These facts have a positive effect on the storage cost value as well as the overall level of capital tied up in inventory. Due to the lower optimal order size, the order cycle is also shorter, thus increasing the inventory turnover rate, which is a significant indicator of the supply process being managed efficiently.
Please, see the attachment as well.

Round 2
Reviewer 3 Report
The authors have addressed my comments satisfactory.